# Remote Sensing of Daytime Water Leaving Reflectances of Oceans and Large Inland Lakes from EPIC onboard the DSCOVR Spacecraft at Lagrange-1 Point

**DOI:** 10.3390/s19051243

**Published:** 2019-03-12

**Authors:** Bo-Cai Gao, Rong-Rong Li, Yuekui Yang

**Affiliations:** 1Remote Sensing Division, Naval Research Laboratory, Washington, DC 20375 USA; rong-rong.li@nrl.navy.mil; 2NASA Goddard Space Flight Center, Greenbelt, MD 20771, USA; Yuekui.Yang-1@nasa.gov

**Keywords:** remote sensing, sensors, ocean color, algae, turbid water

## Abstract

The NASA’s Earth Polychromatic Imaging Camera (EPIC) on board the Deep Space Climate Observatory (DSCOVR) satellite has been making multiple observations of the entire sunlit Earth in a given day from the Sun-Earth Largangian L1 point since the summer of 2015. EPIC contains 10 narrow channels in the 317–780 nm solar spectral range. The data acquired with EPIC have already been used in a variety of scientific investigations, including the study of the global ozone levels, aerosol index and aerosol optical depth, UV reflectivity of clouds over land and ocean, cloud height over land and ocean, and vegetation indices. In this article, we report that EPIC data, particularly for the data measured with narrow channels centered near 443, 551, and 680 nm, can also have important applications in remote sensing of ocean color in different geographical regions. We have modified a version of a multi-channel atmospheric correction algorithm for Moderate Resolution Imaging SpectroRadiometer (MODIS) ocean color applications and adapted the algorithm for processing EPIC data. We present three case studies on water leaving reflectance retrievals from EPIC data acquired over a large turbid river, inland lakes, and oceans. We conclude that a future ocean color instrument on board a satellite at the L1 point, which provides continuous view of the full sunlit disk of the Earth, will complement and extend ocean color observations with the low Earth observing polar orbital and geostationary satellite instruments in both the spatial and time domains.

## 1. Introduction

The NASA’s Earth Polychromatic Imaging Camera (EPIC) [1] on board the Deep Space Climate Observatory (DSCOVR) satellite has been making multiple observations of the entire sunlit Earth in a given day since the summer of 2015. Historically, the DSCOVR mission evolved from the late 1990s NASA intended Triana Mission, which was strongly endorsed by former vice president Albert Gore [2], for climate related research. Triana was a mission designed for deployment into a stable orbit, at approximately 1.5 million kilometers from the Earth in the direction of the Sun. The orbit is known as the Lissajous orbit about the Lagrangian point 1 (L1), which is stable in the sense that the satellite remains near the Sun-Earth line and views the full sunlit disk of the Earth continuously. Triana was proposed to be an exploratory mission to investigate the scientific and technical advantages of L1 for earth observations. The continuous view of the full sunlit disk of Earth was for complementing and extending observations from low Earth orbit (LEO) or geostationary Earth orbit (GEO) satellite instruments. Later on, Triana was renamed to DSCOVR and DSCOVR was launched into space in February of 2015.

The EPIC instrument on board the DSCOVR satellite contains 10 narrow channels in the 317–780 nm solar spectral range. Descriptions of the EPIC instruments have already been given, for examples, by Herman et al. [3], Marshak et al. [1] and Yang et al. [4]. The 10 spectral channels, their full width at half maximum (FWHM), and originally anticipated primary applications are listed in Table 1. The focal plane of the EPIC system is a 2048 × 2048 pixel charge-coupled device (CCD) array. Data for the 10 channels are obtained using a 10-color filter wheel assembly and shutter [4]. The filter transmission functions are given in Herman et al. [3]. The signal to noise ratio (SNR) for the 10 channels is ~200:1 [5]. Over bright surfaces, e.g., ice and snow, or thick clouds, the maximum SNR can achieve 290:1 [3].

From Table 1, it is seen that the observation of ocean color is not listed as a primary objective. Through visual examination of EPIC images posted on public websites and detailed scientific analysis of limited number of EPIC data sets, we have found that the EPIC data can have an additional application, i.e., remote sensing of ocean color.

In this article, we report that EPIC data, particularly for the data measured with narrow channels centered near 443, 551, and 680 nm, can have important applications in remote sensing of ocean color. We have modified a version of a multi-channel atmospheric correction algorithm for Moderate Resolution Imaging SpectroRadiometer (MODIS) [6,7] ocean color applications [8] and adapted the algorithm for processing EPIC data. The data and methods are described in Section 2. Three case studies on water leaving reflectance retrievals over a large turbid river, inland lakes, and oceans are shown in Section 3. Discussions are present in Section 4. Finally, a summary is given in Section 5.

## 2. Data and Methods

As listed in Table 1, EPIC has 10 channels in the solar spectral range between 317 and 780 nm. The ocean color information is mostly contained in images of the blue (443 nm), green (551 nm), and red (680 nm) channels. The other channels, in general, do not contain much color information about water surfaces. We use an EPIC data set acquired on 13 March 2017 at UTC 1442 to illustrate the properties of EPIC data. Figure 1a shows the sunlit portions of the full disk color image processed from the satellite Level 1B “apparent reflectance images” of three EPIC channels (Red: 680 nm; Green: 551 nm; Blue: 443 nm). At the exact nadir looking direction, the L1B data’s spatial resolution (pixel size) is approximately 8 km. For the off nadir looking pixels, such as those near the edge of the sunlit disk, the pixel sizes on the ground are coarser than 8 km.

The apparent reflectance *ρ** at the satellite for a given channel is traditionally defined [9,10] as:(1)$$\rho *=\frac{\pi L}{\mu_{0}E_{0}}$$
where *L* is the radiance measured by the satellite, *μ*_0_ the cosine of solar zenith angle, and *E*_0_ the extra-terrestrial solar flux. The clear water surfaces appear blue in Figure 1a. Clouds are white. However, most land surfaces appear in the distorted brownish color because the widths of the three channels are too narrow and these channels do not contain the full spectral information for the proper generation of natural-looking color images unless going through a special data processing procedure [3].

Figure 1b shows the black and white image processed from the apparent reflectances of the 764-nm channel, which is located within the strong atmospheric oxygen-A band absorption region. When generating this image, we set the pixels with *ρ** (764 nm) values smaller than 0.05 to be zero. We also set the pixels with *ρ** (764 nm) values greater than 1 to be 1. Clouds and land surfaces are bright in the Figure 1b image, while the blue colored water surfaces in Figure 1a are completely dark. 

The shapes of the blue features in Figure 1a are approximately the same as those of black features in Figure 1b. This indicates that the lower end *ρ** (764 nm) threshold value of 0.05 is appropriate for the separation of water surfaces from land surfaces and clouds. In order to further illustrate this point, we show in Figure 1c an enlarged portion of Figure 1a color image, but just covers the southern portion of S. America. The smaller area within the red-colored lines in Figure 1c contains the very turbid La Plata River, which is difficult to see. Figure 1d is the 764-nm O_2_-A channel image covering the same area as that of Figure 1c. The La Plata River in Figure 1d within the red-colored lines is picked up quite well. Figure 1e shows apparent reflectances as a function of wavelength for a pixel over the turbid river water (the blue line marked with ‘+’) and over a clear land surface area (the red line marked with ‘*’). For the river pixel, because of surface liquid water absorption at 764 nm and the strong atmospheric oxygen absorption at the same wavelength [11,12], the *ρ** (764 nm) value is less than 0.05. Land pixels are generally brighter than water pixels, and the *ρ** (764 nm) values of land pixels are generally greater than 0.05, based on our detailed study of this EPIC data set and a number of other data sets acquired over different days and seasons.

As illustrated in Figure 1e, the multi-channel apparent reflectances as a function of wavelengths for land and water pixels are affected by atmospheric molecular and aerosol scattering effects and atmospheric gas absorption effects, including ozone and oxygen. These atmospheric effects need to be removed in order to obtain land surface reflectances and water leaving reflectances from EPIC data. Previously, we developed a multi-channel atmospheric correction algorithm for MODIS ocean color applications [8]. In the MODIS algorithm, atmospheric aerosol models and optical depths are derived on the pixel by pixel basis with a spectrum-matching technique using MODIS channels centered near 865, 1240, 1640, and 2130 nm. We have modified the MODIS version of the algorithm for processing EPIC data. The modifications include the generation of atmospheric Rayleigh and aerosol scattering tables and gas absorption tables appropriate for EPIC channels. Because EPIC does not have channels centered at wavelengths longer than 800 nm, we do not retrieve aerosol information from EPIC data. Instead, we derive surface reflectances by assuming climatological aerosol models and optical depths from EPIC data acquired on very clear days.

In brief summary, we have developed an empirical but very effective technique for separating water pixels from land and cloud pixels using the 764-nm channel. Compared to using an existing surface type dataset, this technique also avoids the known geolocation accuracy issue in EPIC L1B data due to the rotation of the Earth during the imaging time frame [4]. We have also adapted a previously developed MODIS version of multi-channel atmospheric correction algorithm for processing EPIC data. Examples of water pixel masking and atmospheric corrections are present in the section below.

## 3. Results

We have selected three cases for the demonstration of water leaving reflectance retrievals from EPIC data. The first case is for the scene containing the very turbid La Plata River (see Figure 1) in South America. The second case is for a scene containing all the five large lakes in the Great Lake area in North America. The third case is for scenes containing highly productive Caspian Sea, Black Sea, and Azov Sea in Europe.

### 3.1. La Plata River, South America, March 2017

La Plata River is located in South America. Near the mouth of the river to the Atlantic Ocean, the water is very turbid and often appears in brownish color in satellite images, such as those acquired with the NASA Terra and Aqua MODIS instruments. Because the present ocean color community adopted atmospheric correction algorithms were mainly designed for processing the clear case 1 blue ocean waters, these algorithms generally do not produce reliable water leaving reflectances over turbid case 2 waters as well as over waters with chlorophyll concentration greater than about 10 mg/m^3^ [13]. We selected the EPIC data set acquired on March 13, 2017 at UTC 1442 in the present study. The full disk RGB image and the 764-nm O_2_-A channel image have already been shown in Figure 1a,b, respectively. We have made atmospheric corrections to the EPIC data set for three scenarios: Rayleigh only (no aerosol), Rayleigh plus a climatologically averaged continental aerosol model with an optical depth of 0.1 at 550 nm, and Rayleigh plus a climatologically averaged maritime aerosol model with the same optical depth of 0.1 at 550 nm.

Figure 2a is the full disk color image processed from the atmosphere-corrected data of the three EPIC channels (Red: 680 nm; Green: 551 nm; Blue: 443 nm) assuming the maritime aerosol model with an optical depth of 0.1 at 550 nm. Figure 2b is the enlarged portions of the color image for southern portions of South America. Within the red color outlined rectangle, La Plata River is still difficult to see. Figure 2c is the color image for the same portions of South America with land surfaces masked out using the technique described in Section 2. The turbid La Plata River and nearby clear ocean surfaces are shown quite well. The inlet in (c) shows an enlarged portion of La Plata River. Figure 2d shows the spectral reflectances as a function of wavelength for one turbid river water pixel located at the center of the small rectangle outlined in Figure 2c. The line marked with ‘+’ is the top of atmosphere apparent reflectance curve. The lines marked with ‘*’, diamond, and triangle are the retrieved water leaving reflectances for the Rayleigh only, Rayleigh plus rural aerosol model, and Rayleigh plus maritime model corrections, respectively. The shapes and the magnitude of the two curves for the rural aerosol model and maritime model are very similar—indicating that the exact aerosol model used in atmospheric correction for the bright turbid water pixel is not critically important. However, by careful studies of all the four curves in Figure 4d, it is seen that an accurate Rayleigh correction is very important, because the Rayleigh correction removes the largest amounts of scattering effects in the UV (388 nm), blue (443 nm), and green (551 nm) channels.

### 3.2. Great Lakes, North America, September 2017

Increases in the frequency and magnitude of harmful algae blooms (HABs) over the Great Lake region in North America in recent years have caused serious threats to the fresh and marine aquatic ecosystems and public health risks [14,15]. Satellite remote sensing can help to assess the water quality of the Great Lakes [16]. Because of frequent cloud contamination, it is not easy to find an EPIC scene that is clear over all the five major lakes, i.e., Lake Erie, Lake Michigan, Lake Ontario, Lake Huron, and Lake Superior. After extensive search, we have found one great EPIC scene that was clear over all the five major lakes. The EPIC data set was acquired on 24 September 2017 at UTC 1649.

Figure 3a is the full disk color image processed from the atmosphere-corrected data of the three EPIC channels (Red: 680 nm; Green: 551 nm; Blue: 443 nm) assuming the maritime aerosol model with an optical depth of 0.1 at 550 nm. Figure 3b is the enlarged portion of the color image covering the five large lakes. Figure 3c is the color image covering the same portion of the Great Lake area after masking out the land surfaces and clouds. The left portion of Lake Erie has ‘green’ color because of a major harmful algal bloom event there at the time of EPIC data acquisition [17,18]. Figure 3d shows the spectral reflectances as a function of wavelength for one chlorophyll-rich water pixel in Lake Erie and located exactly at the cross-hair point in Figure 3c. The line marked with ‘+’ is the top of atmosphere apparent reflectance curve. The lines marked with ‘*’, diamond, and triangle are the retrieved water leaving reflectances for the Rayleigh only, Rayleigh plus rural aerosol model, and Rayleigh plus maritime model corrections, respectively. The shape and the magnitude of the two curves for the rural aerosol model and maritime model are very similar, which indicates again that the exact aerosol model used in atmospheric correction for the bright and chlorophyll-rich water pixel is not very important. It is also noted that the green channel (551 nm) reflectance values after atmospheric corrections are larger than those of blue channel (443 nm) and red channel (680 nm). Therefore, we properly recovered the green color of the chlorophyll-rich pixels after atmospheric corrections.

Figure 3e is a true Terra MODIS color image covering approximately the same surface area as that of Figure 3c. The MODIS image with a spatial resolution of about 1 km was downloaded from the NASA MODIS web site. No spatial registration between the Figure 3e,c images is made. By comparing the two images, it is seen that the chlorophyll blooming feature in the left part of Lake Erie in the Figure 3e MODIS image is much sharper than that in the Figure 3c EPIC image. The lower spatial resolution EPIC image with pixel sizes of 24 km or larger smeared out the fine water features.

### 3.3. Caspian Sea, Black Sea, and Azov Sea, Europe, June 2017

Caspian Sea, Black Sea, and Azov Sea are located in Europe. From late June to early September in a year, the waters in the areas are usually highly productive, and chlorophyll bloom features can often be observed from satellite images. It is relatively easy to find good EPIC data sets, because the areas are often clear and free of obvious cloud contamination. In the third case study reported here, we selected the EPIC data sets acquired on 26 June 2017 with several clear overpasses over the areas during the daytime.

Figure 4a is the full disk color image processed from the atmosphere-corrected data of the three EPIC channels (Red: 680 nm; Green: 551 nm; Blue: 443 nm) assuming the maritime aerosol model with an optical depth of 0.1 at 550 nm. The EPIC data was acquired at UTC 1112. Figure 4b is the magnified portion covering the Caspian Sea, Black Sea, Azov Sea, Persian Gulf, Red Sea, and Mediterranean Sea. Figure 4c is the color image covering the same area as that of Figure 4b but with land surfaces and clouds masked out. Figure 4d shows the spectral reflectances as a function of wavelength for two chlorophyll-rich pixels—one located in the upper right portion of Caspian Sea and the other at the center of Azov Sea, one clear water pixel in eastern portion of Black Sea, and one clear water pixel in eastern part of Mediterranean Sea. It is seen that the green channel (551 nm) reflectance value for the Caspian Sea pixel is almost twice as large as that of the Azov Sea pixel. Figure 4c,d demonstrate that we are able to retrieve properly the color of water surfaces after performing the atmospheric corrections.

Figure 5a–h shows eight sunlit full disk color images processed from atmosphere corrected EPIC channel (Red: 680 nm; Green: 551 nm; Blue: 443 nm) images acquired on 26 June 2017 at UTC 0545, 0650, 0756, 0901, 1007, 1112, 1217, and 1323, respectively. The corrections were made by assuming a climatologically averaged maritime aerosol model with an optical depth of 0.1 at 550 nm for all the scenes. The locations of the Caspian Sea in these images are marked with red arrows. These images demonstrate that it is possible to have multi-views of the same area, such as Caspian Sea, with EPIC at the L1 point with a time interval of approximately 1 h and 5 min from the local sunrise to local sunset in a day. Figure 6a–h corresponds to Figure 5a–h, respectively, but covering much smaller areas consisting of Caspian Sea, Black Sea, and Azov Sea and with land and cloud features masked out. The spatial distributions of chlorophyll blooming features in the upper right portions of Caspian Sea and Azov Sea can vary rapidly with time. The multiple view of the areas in a day allows, in principle, the detection of rapid changes of chlorophyll bloom patterns over the areas. 

Because the EPIC pixel sizes on the ground change with local view zenith angles and spatial registration errors associated with different channels in a given EPIC scene, we have not tried to make spatial registrations among the eight images in Figure 6a–h to a common latitude and longitude grid for quantitative inter-comparisons among these images. However, the eight images in Figure 6a–h have demonstrated the capability of acquiring multiple views of a given area within a day with EPIC at the L1 point. Such capability is lacking with the present low earth orbiting polar platform satellite instruments (e.g., MODIS, VIIRS).

## 4. Discussion

At the L1 vantage point, EPIC views the entire sunlit side of the Earth continuously and provides new opportunity for ocean color studies. As mentioned before, the EPIC pixel size at nadir is about 8 km, which is much coarser than the state-of-the-art ocean color sensors, e.g., Visible Infrared Imaging Radiometer Suite (VIIRS) [19]. Data from the L1 point with higher resolution would be much desirable. Resolution can be determined [20] by:*R* ≈ 1.22 *λ S*/*D*(2)
where *R* is the detector’s spatial resolution, *λ* the channel wavelength, *S* the distance between the sensor and the surface, and *D* the aperture diameter, which is 30 cm for EPIC [5]. At the L1 point (*S* = 1.5 million km), the current EPIC instrument can support a 3 km pixel size at the visible spectrum, but to achieve that, the CCD array size (currently 2048 × 2048) needs to be increased.

Another factor affecting data quality is the sensor signal to noise ratio (SNR), which is a function of many parameters, including spatial and spectral resolution, sampling time, sensor optics, detector quantum efficiency etc. The 10 EPIC channels, as listed in Table 1, are all very narrow channels with FWHMs smaller than 3 nm with designed SNR ~200 [5]. These channels were not originally designed for remote sensing of ocean color. The ocean color channels, M1–M7, implemented on the VIIRS [19] instrument currently flying on the Suomi National Polar-orbiting Partnership spacecraft, are much wider with FWHMs of approximately 20 nm. If channels similar to M1–M7 are implemented onto an EPIC-like instrument at the L1 point and keep the same scan rate as that for EPIC, with the state of the art sensor technology and an aperture size of ~100 cm, it is possible to obtain remote sensing data at a spatial resolution of ~1 km or finer with SNR similar to or better than the current EPIC instrument. Such an EPIC-like instrument, if built, would permit full sunlit earth disk observations of ocean color from sunrise to sunset with a repeating cycle of approximately 1 h.

## 5. Summary

The NASA’s EPIC instrument on board the DSCOVR satellite has been making multiple observations of the entire sunlit Earth in a given day from the Sun-Earth Largangian L1 point since the summer of 2015. The data acquired with EPIC have already been used for the study of the global ozone levels, aerosol index and aerosol optical depth, UV reflectivity of clouds over land and ocean, cloud height over land and ocean, and vegetation indices. Through analysis of EPIC data, we have found that the data acquired with narrow channels centered near 443, 551, and 680 nm, can also have important applications in remote sensing of ocean color. We have modified a version of a multi-channel atmospheric correction algorithm for MODIS ocean color applications and adapted the algorithm for processing EPIC data. We have shown three case studies on water leaving reflectance retrievals from EPIC data acquired over a large turbid river, inland lakes, and oceans. We conclude that a future ocean color instrument on board a satellite at the L1 point, which provides continuous view of the full sunlit disk of the earth, will complement and extend ocean color observations with the low earth observing polar orbital and geostationary satellite instruments in both the spatial and time domains.

## Figures and Tables

**Figure 1 sensors-19-01243-f001:**
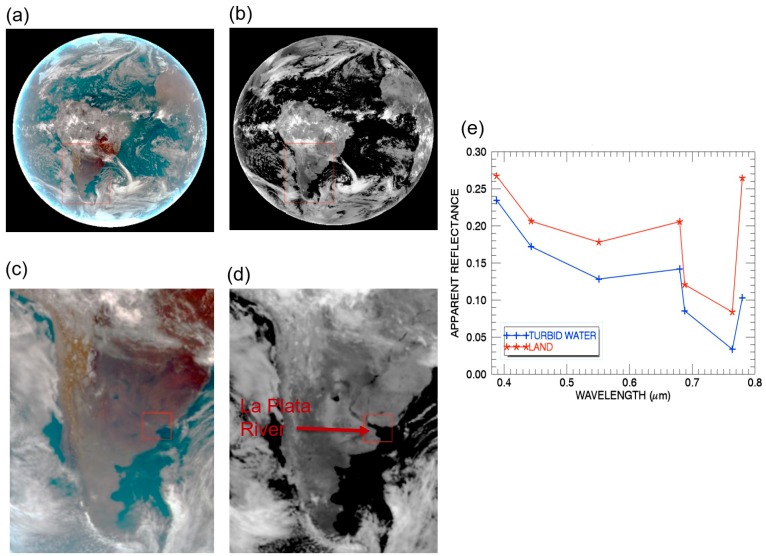
(**a**) The sunlit full disk color image processed from three EPIC channel (Red: 680 nm; Green: 551 nm; Blue: 443 nm) images acquired on 13 March 2013 at UTC 1442; (**b**) The full disk 764-nm channel image; (**c**) The portions of color image over southern part of S. America; (**d**) The portions of the 764-nm image over southern part of S. America; (**e**) Apparent reflectances as a function of wavelength for a turbid river water pixel (blue line marked with ‘+’) and a land pixel (red line marked with ‘*’). Pixels are selected in the red box marked in Panel (C) and (D).

**Figure 2 sensors-19-01243-f002:**
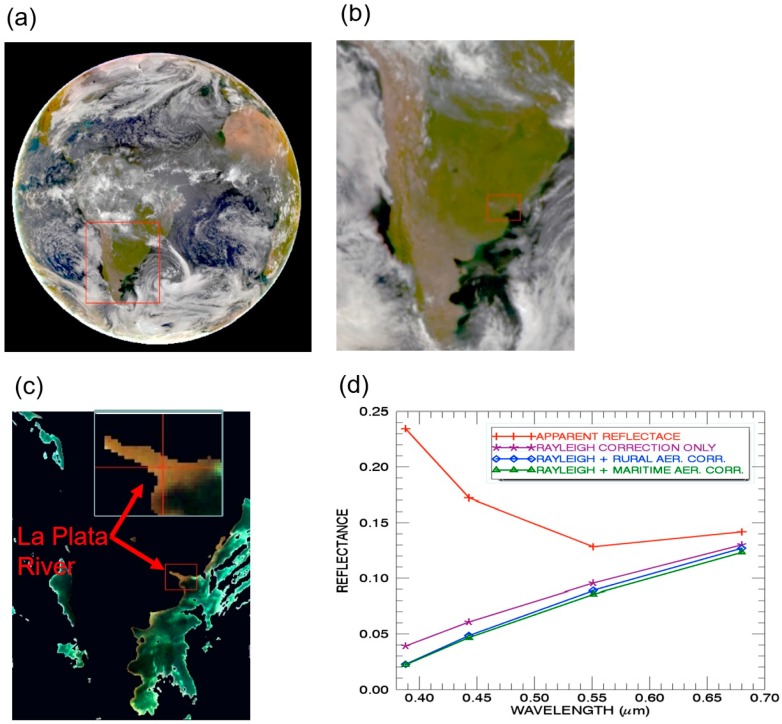
(**a**) The sunlit full disk color image processed from three atmosphere corrected EPIC channel (Red: 680 nm; Green: 551 nm; Blue: 443 nm) images acquired on 13 March 2017 at UTC 1442; (**b**) The portions of color image over S. America; (**c**) The portions of color image of water surfaces over S. America after masking out land and cloud pixels; (**d**) Spectral reflectances as a function of wavelength for one turbid river water pixel located at the center of the small rectangle outlined in (**c**). In (**d**), the line marked with ‘+’ is the top of atmosphere apparent reflectance curve. The lines marked with ‘*’, diamond, and triangle are the retrieved water leaving reflectances for the Rayleigh only, Rayleigh plus rural aerosol model, and Rayleigh plus maritime model corrections, respectively. The inlet in (**c**) shows an enlarged portion of La Plata River.

**Figure 3 sensors-19-01243-f003:**
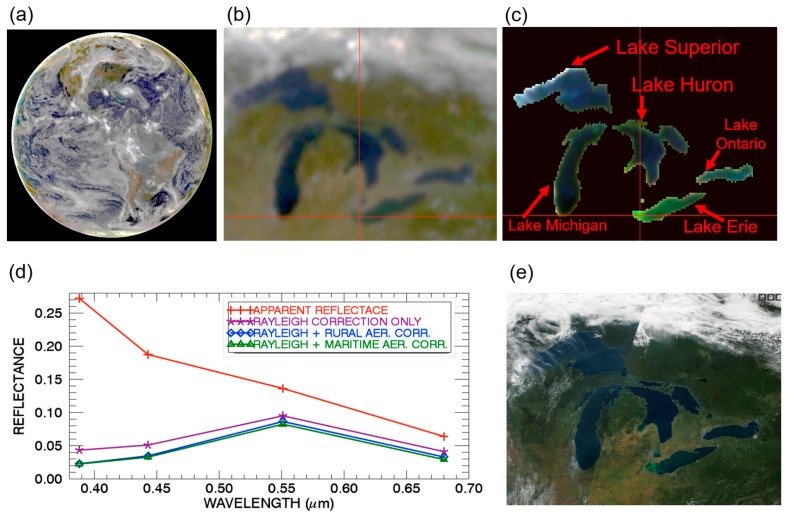
(**a**) The sunlit full disk color image processed from three atmosphere corrected EPIC channel (Red: 680 nm; Green: 551 nm; Blue: 443 nm) images acquired on 24 September 2017 at UTC 1649; (**b**) The portion of color image over Great Lake region; (**c**) The portion of color image of water surfaces over Great Lakes after masking out land and cloud pixels; (**d**) Spectral reflectances as a function of wavelength for one chlorophyll-rich water pixel located in the left part of Lake Erie. (**e**) The 1-km resolution true color Terra MODIS image over the Great Lake area. In (**d**), the line marked with ‘+’ is the top of atmosphere apparent reflectance curve. The lines marked with ‘*’, diamond, and triangle are the retrieved water leaving reflectances for the Rayleigh only, Rayleigh plus rural aerosol model, and Rayleigh plus maritime model corrections, respectively.

**Figure 4 sensors-19-01243-f004:**
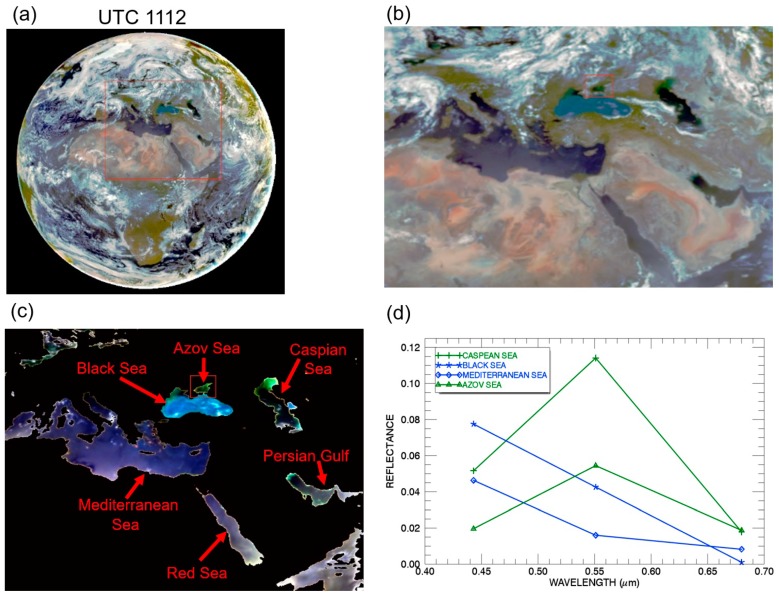
(**a**) The sunlit full disk color image processed from three atmosphere corrected EPIC channel (Red: 680 nm; Green: 551 nm; Blue: 443 nm) images acquired on 26 June 2017 at UTC 1112; (**b**) The portion of image over Caspian Sea, Black Sea, Azov Sea, Persian Gulf, Red Sea, and Mediterranean Sea; (**c**) The portion of image of water surfaces over the same area as that of (**b**), but after masking out the land and cloud pixels; (**d**) Spectral reflectances as a function of wavelength for two chlorophyll-rich water pixels (Green lines) over Caspian Sea and Azov Sea and two blue water pixels (Blue lines) over eastern portions of Black Sea and Mediterranean Sea.

**Figure 5 sensors-19-01243-f005:**
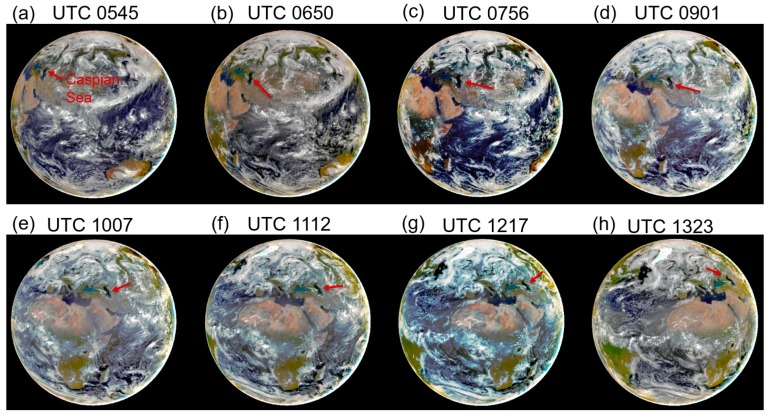
(**a**–**h**) Eight sunlit full disk color images processed from atmosphere corrected EPIC channel (Red: 680 nm; Green: 551 nm; Blue: 443 nm) images acquired on 26 June 2017 at UTC 0545, 0650, 0756, 0901, 1007, 1112, 1217, and 1323, respectively. The spatial locations of the Caspian Sea in these images are pointed to with red arrows.

**Figure 6 sensors-19-01243-f006:**
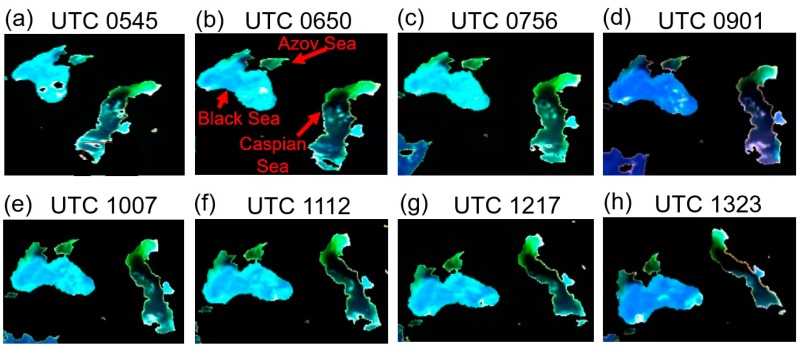
(**a**–**h**) Eight color images processed from atmosphere corrected EPIC channel (Red: 680 nm; Green: 551 nm; Blue: 443 nm) images acquired on 26 June 2017 at UTC 0545, 0650, 0756, 0901, 1007, 1112, 1217, and 1323, respectively, and covering areas consisting of Caspian Sea, Black Sea, and Azov Sea with land and cloud features masked out.

**Table 1 sensors-19-01243-t001:** Specifications of 10 EPIC channels and their primary applications.

Spectral Channels (nm)	FWHM (nm)	Primary Usage
317.5	1	Ozone, SO_2_
325	2	Ozone
340	3	Ozone, Aerosols
388	3	Aerosols, Clouds
443	3	Aerosols
551	3	Aerosols, Vegetation
680	2	Aerosols, Vegetation, Clouds
687.75	0.8	Cloud Height
764	1	Cloud Height
779.5	2	Clouds

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
