# Peer review of "Remote Sensing of Daytime Water Leaving Reflectances of Oceans and Large Inland Lakes from EPIC onboard the DSCOVR Spacecraft at Lagrange-1 Point"

_sensors, 2019, doi:10.3390/s19051243_

Reviewer 1 Report

       A brief summary

The authors suggest using data from the NASA’s Earth Polychromatic Imaging Camera (EPIC) in ocean color remote sensing. The value of those data is that the ones from multiple observations of the entire sunlit Earth in a given day at 10 narrow channels in the range of 317-780 nm. The authors have adapted the MODIS algorithm of atmospheric correction for using that with EPIC data. Examples of the obtained results in different geographical regions are presented.

Brief comments

First of all, it should be noted the novelty and usefulness of the presented work. The ability to observe simultaneously a huge water surface and hourly follow occurring changes, opens up broad prospects for future research on the processes determining these changes, monitoring and forecast. But the goal of ocean color remote sensing not only the reflectance, it includes also the retrieval of the bio-optical parameters such as chlorophyll concentration, the color dissolved organic matter and suspended particle content, the absorption and backscattering seawater coefficients and so on. In this aspect, the accuracy of determining the spectral values of the remote sensing reflectance from EPIC data is crucial. Unfortunately, the authors practically do not consider this problem. But it should have done, given in particular that a highly simplified method of atmospheric correction is used (not current data on atmospheric aerosol but the climatological aerosol models). This could be done, at least, by direct comparison between the concurrent spectral values of water reflectance retrieved from EPIC and MODIS (or VIIRS) data.

Specific comments

1. I would recommend to add the useful additional information for each EPIC channel (such as the signal/noise ratio and other).

             2. Line 77 “…natural-looking color images” – the conventional term used in satellite ocean color is “true color image”.

Reviewer 2 Report

Thank you for the opportunity to review this paper.  This paper presents the potential application of the EPIC sensor in deriving water leaving radiance and makes  and argument for the development of such a sensor in a similar orbit with the channels similar to ViiRS.  The manuscript is relevant to the journal Sensors and should be published after minor corrections.

General Corrections and Questions.

Many of the methods and results are presented in a repetitive manner.  I was wondering if some of the repeated results and methods could be presented in a table or a synthesized text at the beginning of each section.  Why weren't spatial registrations conducted (Line 262).  That would have been useful in trying to match up derived chlorophyll measurements with in situ measurements, easily acquired in places such as Lake Erie.  Further to this, I am wondering why chlorophyll algorithms weren't applied to the data and validated against in in situ measurements.  are poor match ups expected given the current channel configuration?  Doing this would make it a much more robust paper providing more information regarding the feasibility of using this sensors for biophysical studies.

Minor Corrections

Title

The title could be improved.  Are you actually trying to detect water leaving reflectance? The word sunlit seems a little confusing or redundant and makes me think this is a paper about glint detection correction.  How about a title something like this..."Assessing capabilities of EPIC for detecting water leaving reflectances and chlorophyll retrieval" 

Abstract

Lines 16-17 - should index be indices throughout the sentence because you are referring to several indices?

Introduction

Line 46 - Examples should be example

Line 48 - Examinations should be examination

Line 49 - rewrite to say "posted on public websites"

Data and Methods

Line 71 - the symbol for apparent reflectance (rho?) was missing in my PDF printout.  

Line 73 - same here, mu is missing in my PDF printout

Line 80 - Oxygen should not be capitalized

Line 81 and 82 - rho symbols are missing?

Line 94 - Inconsistent referencing.  These should be [10,11]

Results

Try to reduce the repetition throughout the results section...for example, "assuming the maritime aerosol model with an optical depth of 0.1 at 550nm"  Does this have to be repeated throughout the results or can this be simply indicated in the methods section as this is how all the images were processed.

Line 177 - contaminations should be contamination

Line 187 - chlorophyll bloom should be harmful algal blooms.  Please provide some references regarding this event or HABs in Lake Erie in general.

https://earthobservatory.nasa.gov/images/91017/bloom-persists-in-lake-erie

https://www.sciencedirect.com/science/article/pii/S0380133018301874

Line 204 - Even though EPIC smeared out the fine details of the bloom event I am not sure that leaving this sentence in will help make your case for the use of EPIC imagery.  I see that you try to make a case for EPIC, or a similar sensor which has higher resolution channels, in the discussion.  This argument against EPIC seems out of place.

Line 224 - Again, there is a lot of repetition in this paragraph and some of the Greek characters seem to be missing.  Not sure if this is just font compatibility on my printer?

Line 259 - multi should be multiple

Line 260 - allow should be allows

Line 265 - of multi-view should be of acquiring multiple views

Discussion

Well written

Summary

Line 308 - Here you discus water reflectance retrievals.  Maybe this word should be used in the title.

The figures and captions look fine.  Overall a great and interesting study of the use of the EPIC sensor and a strong justification for a newer high resolution sensor at the same orbit as EPIC.
